# Time in Range in Children with Type 1 Diabetes before and during a Diabetes Camp—A Ceiling Effect?

**DOI:** 10.3390/children9121951

**Published:** 2022-12-12

**Authors:** Katrin Nagl, Ina Bozic, Gabriele Berger, Martin Tauschmann, Nicole Blauensteiner, Katharina Weimann, Julia K. Mader, Birgit Rami-Merhar

**Affiliations:** 1Department of Pediatrics and Adolescent Medicine, Medical University Vienna, 1090 Vienna, Austria; 2Diabetes Outpatient Clinic, OEGK Health Care Centre Vienna Floridsdorf, 1090 Vienna, Austria; 3Division of Endocrinology & Diabetology, Department of Internal Medicine, Medical University of Graz, 8036 Graz, Austria

**Keywords:** type 1 diabetes, CGM, time in range, time below range, time above range, diabetes camp, children, rtCGM, isCGM

## Abstract

Background: The aim of this study was to assess and compare the time in range (TIR) of children with type 1 diabetes (T1D) before and during a diabetes summer camp using different therapy modalities. Methods: A retrospective analysis of continuous glucose monitoring (CGM) data collected from 26 children with T1D (mean age: 11.0 ± 1.4 years; 62% female; 62% on insulin pump; Hb1Ac 7.3 ± 0.8% (56.3 ± 8.7 mmol/mol) before and during a 14-day summer camp. CGM methods: 50% intermittently scanned CGM (isCGM) and 50% real-time CGM (rtCGM). No child was using a hybrid closed loop system. Results: Mean TIR during camp was significantly higher than before camp ((67.0 ± 10.7%) vs. 58.2% ± 17.4%, *p* = 0.004). There was a significant reduction in time above range (TAR) (*p* = 0.001) and increase in time below range (TBR) (*p* < 0.001), Children using isCGM showed a more pronounced improvement in TIR during camp compared to rtCGM-users (*p* = 0.025). The increase in TIR strongly correlated with numbers of scans per day in isCGM-users (r = 0.751, *p* = 0.003). Compared to isCGM-users, rtCGM-users showed significantly less TBR. The TIR target was met by 30.8% of participants during camp. Conclusion: Glycemic control improved significantly during the camp. However, on average, the therapy goal (TIR > 70%) could not be achieved despite great professional effort.

## 1. Introduction

Continuous subcutaneous glucose monitoring (CGM) facilitates diabetes management and helps improve glycemic control [1,2,3] in people living with type 1 diabetes (T1D) [4,5]. CGM systems [6] might even play the main role in improving glycemic control, while the method of insulin application could be of secondary importance (i.e., insulin pen vs. insulin pump) [2,7]. However, the distinction between real-time CGM systems (rtCGM) and intermittently scanned CGM systems (isCGM) is important. As opposed to isCGM, rtCGM provide continuous information about glycemia and warn about impending hypoglycemic or hyperglycemic events and glycemic excursions [8]. 

The ALERTT1 study showed that a switch from isCGM to rtCGM improved TIR in adults significantly after 6 months [9], while under sub-optimal control, such a switch leads to benefits in terms of hypoglycemia prevention but no significant reduction in hyperglycemia [10]. 

Direct comparisons between the effect of rtCGM vs. isCGM use on glycemic control are scarce [9,10,11,12,13], especially in pediatric cohorts [14]. A disadvantage of rtCGM instead of isCGM might be the complexity of application [8] and that the accuracy of some rtCGM sensors depends on the precision of twice daily calibrations [15]. Furthermore, frequent rtCGM alarms can eventually lead to alarm fatigue [16] and thus to users’ rejection of the system as a whole, which, in turn, may increase preference for isCGM. 

The application of therapy modalities can be examined much better at the controlled, but real-life environment of a diabetes camp. To date, there is no pediatric study directly comparing CGM therapy modalities in a diabetes camp setting. Nor is there a study that dealt more closely with the achievement of glycemic targets (time in range (TIR), time above range (TAR) and time below range (TBR)), during a diabetes camp, paying particular attention to the type of sensor used. This seemed particularly interesting to us, because there are clear differences when using isCGM and rtCGM, such as: self-determined or caregiver-determined checking of sensor values (scanning) for isCGM systems and predefined target range-triggered alarms of values outside the target range for rtCGM systems. In addition, we were interested in whether the scanning behavior of isCGM users changed at the camp compared to the period before camp and whether this change was associated with glycemic control. This is particularly interesting as there are currently no concrete recommendations for the daily number of scans per day in children using an isCGM.

Therefore, we retrospectively compared glycemic control [6] in children with T1D, more specifically TIR, TAR and TBR before and during diabetes camp, focusing on different therapy modalities, such as rtCGM vs. isCGM with special attention to the individual daily scan frequency. 

## 2. Materials and Methods

As described in our previous publication [17], data were gathered during a 2-week summer camp for diabetes education and recreation for children with T1D aged 8 to 12 years in July 2019. The camp aims to connect children with T1D and improve self-confidence and independence regarding their diabetes management [17]. The team of supervisors consists of medical doctors, diabetologists, nurses, medical students, dieticians, and pedagogical staff, who were constantly present. 

The main study “Diabetes Knowledge and Skills in Children with Type 1 Diabetes before and after participation in a diabetes camp” was registered and approved by the ethics board of the Medical University of Vienna (EK-Nr. 1394/2018, Reg. Nr. DRKS00020415). Only data from children using CGM systems before and during camp was analyzed in this sub-study. All children continued to use the insulin regimen and CGM system they had already been using at home. CGM systems used by the participants were: Medtronic Enlite (ENL), Dexcom G6 (DEX), both rtCGM-systems and Abbott Freestyle Libre (FSL), an isCGM-system. 

Inclusion criteria were documented sensor use for at least 70% of the time over at least 14 consecutive days during a period of 30 days prior to the camp and documented sensor use for more than 70% of the time during camp. This resulted in 26 participants. See Appendix A Appendix A for further details. 

Details on the procedures during the camp were already published [17]. At camp, there were three to four planned meals during the day. For every child individually, carbohydrates were counted using scales and assisted by dietologists. Children did not have access to carbohydrates in between meals unless they needed treatment for (impending) hypoglycemia. Every day, there was a structured daily program, which was essentially the same for everyone. This included a wide range of activities, such as physical exercises, swimming, ball games and more.

Insulin dose adjustments (such as insulin/carbohydrate ratios, insulin sensitivity or basal insulin dosage) were performed by diabetologists. The insulin dosage was continuously adapted as required with consideration of current glucose trends, previous values and planned activities and meals.

CGM was routinely performed throughout the camp. During the day sensor readings were checked by children together with medical staff several times a day with additional fingerprick measurements before every meal, exercise activity, before bedtime, in case of signs of hypo- or hyperglycemia and signal loss [17]. During night (22:00–07:00), sensor readings were checked twice every third hour by medical staff. Corrections with insulin or carbohydrates, as appropriate, were performed under supervision day and night. All obtained blood glucose values and amount of insulin and carbohydrates were documented in a paper-chart by medical staff.

During the last two camp days, all data (two camp weeks and 30 days prior camp) were downloaded from all devices (CGM-systems and pumps) for later analysis.

Data analysis was performed retrospectively using downloaded data (as described above) and evaluation of paper charts. 

Time in range (TIR) (70–180 mg/dl (3.9–10 mmol/L)) (6), time below range (TBR) (<70 mg/dL (3.9 mmol/L)), time above range (TAR) (>180 mg/dL (10 mmol/L)) and time using CGM (%) were calculated for each individual separately and are given as percentages. Additionally, TIR, TBR, TAR and time using CGM were calculated separately for the pre-camp and camp timespan. Time before camp was defined as the time from 13 June 2019 0:00 am to 13 July 2019 4:00 pm. Time during camp was defined as the time from 13 July 4:00 pm to 27 July 12:00 pm. Additionally, for TIR, TBR and TAR daytime as well as nighttime was analyzed separately. Nighttime was defined as the period when children were supposed to be in bed (10.00 pm until 7.00 am). 

Glycemic targets were defined according to the international consensus on TIR [6]: TIR > 70%; TBR < 4% and TAR < 25%. Achievement of these targets was given in percent. For isCGM, daily number of scans before and during camp were calculated.

Individual total daily doses (TDD) of insulin (IU/kg/d) and amount of carbohydrate consumption were extracted from pump data and paper charts and further analyzed regarding percentage of basal insulin. Pre-camp information on insulin dose prescription of pen-users were collected by questionnaires directly prior camp. 

Body mass index (BMI) was calculated as weight in kilograms divided by squared height in meters. In order to adjust for age and sex, BMI standard deviation score (BMI-SDS) values were derived applying the least mean square method (Box-Cox-Transformation by Cole et al.) [18] using age- and sex-specific BMI-reference values based on the World Health Organization (WHO) growth reference for school-aged children and adolescents [19].

Baseline characteristics are given as mean ± standard deviation, median (25th, 75th percentile) or percentages, as appropriate. All statistical analyses were performed using Microsoft Office Excel 2019 or SPSS 26.0. Differences between pen vs. pump users and rtCGM vs. isCGM users were analyzed using chi-square tests for qualitative data. For quantitative data with normal distribution, *t*-tests were used for comparisons between groups. Pre-camp and Camp TIR, TAR, TBR and sensor usage were compared using paired *t*-tests, in case of normal distribution. For quantitative data without normal distribution Mann–Whitney U-Test and Wilcoxon Rank Test were used for comparison, and data are given as median [25th;75th percentile]. Pearson’s correlation coefficients (two-tailed) were used to describe the strength of relationships between the dependent variables (TIR, pre-camp TIR and improvement of TIR) and potential covariates (HbA1c, pre-camp TIR, scans per day, camp TIR). An alpha-level of *p* < 0.05 (two-tailed) was considered statistically significant. 

## 3. Results

Baseline characteristics of patients are displayed in Table 1. Sensor distribution was 50% FSL, 35% ENL and 15% DEX. Insulin pumps were used by 61.5% of children. See Appendix A Appendix A for more details.

A median number of 28.5 [23.5;29.7] days during one month before camp, and 12.5 [11.9;12.6] days during camp were analyzed, with a median sensor usage of 95 [82;99]% before camp and 96 [92;97]% during camp, *p* = 0.093. 

### 3.1. Comparison Pre-Camp to Camp

On average, pre-camp TIR significantly improved from 58.2% ± 17.4% to 67.0 ± 10.7% during camp, *p* = 0.004 (Figure 1). There was no significant difference of mean TIR between the first (66.4 ± 12.7%) and second week of camp (67.6 ± 11.4%), *p* = 0.59.

TAR was significantly lower during camp (27.5 ± 10.2%) than before camp (38.7 ± 16.9, *p* < 0.001), while TBR significantly increased from 3.2 ± 2.3 to 5.5 ± 3.0% (*p* < 0.001) (Table 2). 

### 3.2. Comparison of Day- and Nighttime

Significant pre-camp to camp differences for TIR, TAR, TBR were also seen, when daytime and nighttime were analyzed separately (Table 2). During camp, overall nighttime TIR (74.1 ± 10.3%) was higher than daytime TIR (62.5 ± 12.7%), *p* < 0.001). While before camp, the target for TAR (<25%) was not met—neither during the day nor during the night, during camp, the TAR target was met during night. 

At camp, the target for TBR (<4%), was only achieved during the day, but not during night, where TBR even increased to 7.6 ± 5.0% (Table 2).

### 3.3. Insulin Dose Adjustments

In comparison to pre-camp, total daily insulin (TDD) dosage was reduced by 18.3 ± 11.0% during camp. There was no difference in the extent of reduction between the pen and the pump group, or between isCGM and rtCGM users. Percentage of basal insulin was slightly increased during camp (Table 2). 

### 3.4. Comparison between Insulin Pen and Pump Users

Comparing changes in TIR, TAR and TBR pre-camp to camp between pen and pump users resulted in no statistical differences (Appendix A Appendix A).

### 3.5. Comparison between rtCGM and isCGM Users

In comparison to rtCGM-users, isCGM showed a significantly more pronounced pre-camp to camp increase in TIR (+15.0 ± 15.9% vs. +2.8 ± 9.0%, *p* = 0.025) and decrease in TAR (−17.6 ± 16.6% vs. −4.7 ± 9.8%, *p* = 0.024), (Table 2). TBR and TAR before camp were significantly less favorable in isCGM than in rtCGM users. 

Overall, TBR increased both in isCGM and rtCGM users compared to pre-camp. However, TBR was significantly pronounced in the isCGM group, both during the camp and the pre-camp period. Daytime TBR did not differ between isCGM and rtCGM users, neither pre-camp nor during camp. During daytime, absolute differences in TBR between the two groups were small. While there was no significant change in daytime TBR in the isCGM group, the rtCGM group showed a small but statistically significant increase in daytime TBR at camp (1.6 [0.8;2.5]%) compared to the period before the camp (2.9 [1.8;5.3]%), which was nonetheless still within the target range of TBR < 4%. 

Nighttime TBR between the two groups showed a distinct difference. In both groups, there was a clear increase in nocturnal hypoglycemia (nighttime TBR), which was particularly pronounced in the isCGM group. On average, isCGM users spent 11.3 ± 3.9% of the night below the target range, whereas in rtCGM users the TBR-target (<4%) was still met.

Before camp, daytime TAR differed significantly between isCGM and rtCGM users, (51.6 ± 14.2 vs. 31.9 ± 17.3, *p* = 0.004). During camp, there was no significant difference in TAR between the two groups. 

The number of daily isCGM-scans increased significantly pre-camp to camp (13 ± 5 vs. 18 ± 5, *p* = 0.011). 

### 3.6. Associations with Improvement of TIR

In isCGM users the increase in daily scans showed a strong, positive association with TIR-increase (r = 0.751, *p* = 0.003). The increase in TIR was particularly more pronounced when pre-camp TIR was low (r = −0.787, *p* < 0.001). 

### 3.7. Targets and Safety

Before camp the only glycemic target, which was met, was TBR, with an overall TBR of 3.2 ± 2.3%. More precisely, pre-camp, the TBR target was only met by rtCGM users, with an average TBR of 1.8 ± 1.3%, compared to 4.5 ± 2.4% TBR in isCGM users. 

During camp, with an average TIR of 71.9 ± 12.2%, TBR of 3.8 ± 2.3% and TAR of 24.3 ± 12.1% glycemic targets were also only met by the rtCGM group. Individually, however, TIR, TBR and TAR targets were met by 30.8%/34.6%/34.6% of all participants during camp, respectively. Among rtCGM users 43% reached the TIR target, while among isCGM users, only 17% were able to reach the TIR target during camp. Only 3 participants were able to achieve all three (TIR, TAR and TBR) targets during the camp, while 11 participants achieved none. During the camp no severe hypoglycemia, no ketoacidosis or any other relevant intercurrent illness requiring medication or treatment at a hospital occurred.

## 4. Discussion

### 4.1. Improvement of Glycemic Control

Overall, TIR improved significantly compared to the reference period before camp. Nonetheless, the consensus targets were achieved only in a third of our patients, despite great commitment and 24/7 effort by an interdisciplinary team. Only three children achieved all three TIR/TAR/TBR targets. Two out of these three participants were still in remission phase, with a diabetes duration of 3 months, and used DEX and insulin pens. However, compared to the ALERTT1 trial, where only 28% of rtCGM and 15% of isCGM users reached the TIR target [9], the results of our camp study are satisfying. 

With a mean HbA1c of 7.3 ± 0.8% the diabetes control of camp participants within the three months before camp had already been good and is comparable to the glycemic control in this age group as evident in the Prospective Diabetes Follow-up registry (DPV) [3,20], which includes an estimated 80% of children with T1D in Germany, Austria and Luxembourg [21]. 

The improvement in glycemic control amounted to a significant increase in TIR and a significant decrease in TAR. Despite a significant reduction in TDD and continual adjustments of the insulin dose, there also was a significant increase in TBR, which was presumably a consequence of an increase in exercise at camp. 

### 4.2. Day and Night

Overall, nighttime was better than daytime glycemic control. Particularly, there was a clear decrease in TAR, comparable to the results of the ALERTT1 trial [9]. While the authors of the ALERTT1 trial contribute this to the setting of alarms, in our camp study this improvement is most likely attributable to the regular controls and administration of correction insulin doses by the camp staff at night, as there was no statistical difference in nighttime TAR between isCGM and rtCGM users. 

In general, more time was spent in hypoglycemia during the night than during the day, which is comparable to the results of the CORRIDA trial that investigated the effects of 4 days of physical activity on glycemic control in adults [12]. This suggests that the insulin reduction carried out at camp was insufficient, despite insulin being reduced by almost 20% on average in line with suggestions for insulin dose adaption at diabetes camps [22,23]. Additionally, TBR was probably easier to control during the day because of the children’s own hypoglycemia awareness and subsequent faster counteraction. 

### 4.3. Comparison of isCGM vs. rtCGM

A major difference in glycemic control was evident between isCGM and rtCGM groups, despite having the same level of professional, multidisciplinary care. Most children with isCGM were able to significantly improve their pre-camp glycemic control during camp. They almost achieved the same high level of TIR observed in children using an rtCGM system. Specifically, this improvement was mostly a result of a reduction in TAR. 

With an average of 18 scans per day at camp, scans were performed almost hourly in the isCGM group. As the increase in daily scans pre-camp to camp was strongly associated with the improvement in TIR, the number of scans seem to have been a key to success in the isCGM group. This is an interesting finding, as the number scans in our study, are much higher than in other studies [9,24,25,26,27]. A Suisse study in children with a T1D reported 6.5–9 scans per day [26], an Italian study reported a consistent number of 10 scans per day over the period of 1 year [25]. The ALERTT1 study [9] in adults reported 11 scans per day. Furthermore, although the average number of daily scans documented pre-camp (13 scans per day) had also been higher than in other studies [9,24,25,26,27], apparently isCGM systems were still sub-optimally scanned at home. Therefore, it might be reasonable to advice patients with isCGM systems to scan devices even more frequently, especially when glycemic fluctuations are likely to be expected. A recent study on children and adolescents with T1D using isCGM showed a U-shaped relationship between daily numbers of scans count and change in glycemic control. In this study, too, the optimum number of isCGM scans was 15–20 per day [28].

However, an essential difference between isCGM and rtCGM systems was found in nighttime TBR with 11.3% compared to 3.9%. 

This result was in line with results from the CORRIDA study, which also reported how differences in rtCGM and isCGM concerning hypoglycemia were most notable during night [12]. 

Although both rtCGM and isCGM systems are covered by the Austrian health insurance, most children participating at the summer camp used the FSL (isCGM) system [17]. A possible explanation for this might be that isCGM are easy to use, do not have to be calibrated and allow to feel more “carefree” due to the lack of disruptive alarms. In addition, some authors also see greater benefits in using isCGM systems with regard to psychological outcomes in adolescents [8].

However, it may be that precisely these alarms, which are often perceived as annoying [29], made the decisive difference in glycemic control. Close supervision and control at camp only led to a significant improvement in children using isCGM while children using an rtCGM-system already had achieved good glycemic control before camp, possibly due to the alarm functions of their systems, and the possibility to use their devices in combination with advanced technologies such as predicted low glucose suspend (PLGS) technology.

### 4.4. Therapy Modalities

Remarkably, there was no difference in the glycemic control between pen or pump users with both groups having a similarly good glycemic control. This finding supports the hypothesis that CGM system use is probably more essential for achieving good glycemic control than the way in which insulin is administered [2]. 

### 4.5. Diabetes Camp Effect

The overall improvement in glycemic control may also be based on a certain peer effect. Together as a group the participating children were possibly more motivated to carry out the diabetes therapy with more effort and care [30]. 

Moreover, the major difference to the situation at home is that at the camp, the children were looked after 24/7. Even during the night, glucose controls were carried out regularly and it was possible to react immediately and appropriately to alarms and glucose values outside the target range. The structured daily routine with regular rounds of measurements before meals and during the night had a positive effect, especially in the group of isCGM users, by increasing their scanning frequency. With almost 18 scans per day, the frequency of scans at the camp was very high. It is possible that a certain peer effect caused the children to additionally perform scans when their peers were scanning for symptoms such as hypoglycemia, even when they themselves were not presenting with such symptoms.

However, of note is that this high level of effort provided by a multidisciplinary diabetes summer camp cannot be carried out by all families at home, especially not during the night.

### 4.6. Limitations

Although this study provides valuable insights into the use of various CGM systems and improvement of glycemic control in children participating in a diabetes camp, there are several limitations. First, the cohort was small due to the inherently limited number of camp participants. In addition, the retrospective design of the study represents a important limitation and that the selection of the camp participants and the choice for the individually applied CGM systems were not the responsibility of the study team. 

Furthermore, although we did have access to the pump and sensor data for one month before the summer camp, we had limited information about parental support, meals, exercise, etc. at home. 

As the group of children using their insulin pump with a PLGS mode was small, comparing children using PLGS with other groups would not have yielded meaningful statistical results. It can therefore not be ruled out that a part of the better performance in terms of hypoglycemia prevention in the rtCGM group was attributable to the use of PLGS. However, even in the group of children who used the combination of a Medtronic 640G pump and Enlite sensors, more than half did not reach the glycemic targets during camp.

Beyond that, we do not have data on glycemic control after camp participation, nor age- and gender matched control groups, who did not participate in a diabetes camp at all.

## 5. Conclusions

In contrast to rtCGM users, isCGM users showed a significant improvement in glycemic control under camp conditions. However, this was probably mainly due to the structured monitoring not only during the night, which partially replaced the missing alarm function, and to suboptimal at-home use of isCGM systems. In isCGM users, there was a clear correlation between the improvement in glycemic control and the increase in scan frequency. The strong increase in TBR among isCGM users shows once again that the management of nocturnal hypoglycemia is a challenge, even or especially in a camp setting and that spot measurements are not sufficient.

On average, during and before camp, TIR was higher in rtCGM than in isCGM users. As glycemic control in the rtCGM group had already been good before camp, it was not further improved, or rather differences did not reach statistical significance. However, even in the rtCGM group, more than half of the patients did not achieve a TIR > 70% during the camp.

Without the use of artificial pancreas or hybrid-closed loop systems, there seems to be a certain ceiling effect under real-life conditions regarding improving glycemic control in children as the glucose targets—on average—were not achieved despite great and proactive effort and 24/7 medical care. 

The fact that even in this well-controlled camp setting among children with T1D with already good diabetes management, glycemic targets could only be reached by a minority, underlines the burden for families with children with T1D and raises questions whether these glycemic targets are achievable, at least during a diabetes camp loaded with varied activities. Several studies have so far shown that artificial pancreas systems represent a benefit for families and children with T1D [31]. These systems should therefore be evaluated in a real-life setting, such as a camp setting, and compared to sensor-augmented pump and pen therapy to determine whether they would enable patients to better reach TIR targets.

## Figures and Tables

**Figure 1 children-09-01951-f001:**
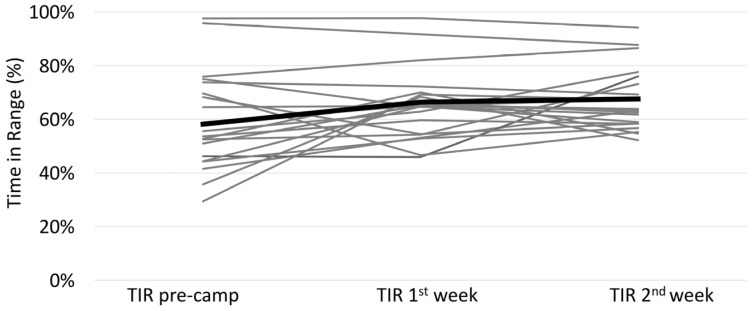
Time in Range (TIR) before and during Camp. Comparison of Mean TIR before camp, during first week and during second week of camp of each individual camp participant (grey). Mean TIR before camp, during first and second week of all camp participants (black, thick line).

**Table 1 children-09-01951-t001:** Baseline Characteristics.

	All	Pen	Pump	*p*-Value	isCGM	rtCGM	*p*-Value
N (%)	26	10 (38.5%)	16 (61.5%)		13 (50%)	13 (50%)	
Age (years)	11.0 ± 1.4	11.4 ± 1.1	10.7 ± 1.5	0.24	11.1 ± 1.2	10.8 ± 1.6	0.58
female/male	10/16	4/6	6/10	1.0	5/8	5/8	1.0
Diabetes duration (years)	4.1 ± 3.1	3.0 ± 3.1	5.0 ± 2.9	0.11	4.2 ± 2.5	4.2 ± 3.6	0.95
Height (cm)	145.9 ± 10.3	146.4 ± 9.8	145.7 ± 10.9	0.87	146.3 ± 8.1	145.5 ± 12.5	0.84
Height-SDS	−0.83 ± 0.97	−0.46 ± 0.83	0.15 ± 1.00	0.12	−0.13 ± 1.16	−0.03 ± 0.79	0.79
Weight (kg)	36.8 ± 7.63	35.4 ± 5.1	37.7 ± 8.9	0.41	38.5 ± 7.13	35.1 ± 8.0	0.265
BMI-SDS	−0.13 ± 0.98	−0.53 ± 0.65	0.13 ± 1.07	0.09	0.17 ± 0.92	−0.42 ± 0.97	0.127
HbA1c (%)	7.3 ± 0.8	7.2 ± 1.1	7.3 ± 0.7	0.66	7.6 ± 0.8	7.0 ± 0.8	0.07
HbA1c (mmol/mol)	56.3 ± 8.7	55.2 ± 12.0	56.3 ± 7.7		59.6 ± 8.7	53.0 ± 8.7	

Qualitative data is presented in total numbers and percentages (%). Chi-Square test was used for comparison. For quantitative data with normal distribution, data is given as mean ± standard deviation, *t*-test was uses for comparison between groups.

**Table 2 children-09-01951-t002:** Glycemic Control before and during Camp.

				Difference between isCGM and rtCGM			Difference Pre-Camp to Camp	
		All		isCGM	rtCGM	*p*-Value	isCGM	rtCGM
	N	26	*p*-Value	13	13		*p*-Value	*p*-Value
Time using CGM (%)	pre-camp	95 [82;99]	0.093	97 [71;100]	94 [90;96]	0.55	0.422	0.463
	camp	96 [92;97]		97 [96;98]	92 [86;96]	0.006		
Time in range (TIR)	pre-camp	58.2 ± 17.4	0.004	47.2 ± 12.2	69.1 ± 14.9	<0.001	0.005	0.288
(%)	camp	67.0 ± 10.7		62.1 ± 6.3	71.9 ± 12.2	0.019		
Time below range	pre-camp	3.2 ± 2.3	<0.001	4.5 ± 2.4	1.8 ± 1.3	0.002	0.006	0.004
(TBR) (%)	camp	5.5 ± 3.0		7.1 ± 2.7	3.8 ± 2.3	0.002		
Time above range	pre-camp	38.7 ± 16.9	0.001	48.3 ± 12.8	29.0 ± 15.0	0.002	0.002	0.109
(TAR) (%)	camp	27.5 ± 10.2		30.7 ± 7.1	24.3 ± 12.1	0.114		
Daytime TIR (%)	pre-camp	55.5 ± 18.4	0.042	44.8 ± 12.9	66.2 ± 17.1	0.001	0.04	0.66
	camp	62.5 ± 12.8		57.4 ± 9.5	67.6 ± 13.9	0.040		
Nighttime TIR (%)	pre-camp	62.5 ± 17.1	<0.001	51.3 ± 13.4	73.8 ± 12.6	<0.001	<0.001	0.038
	camp	74.1 ± 10.3		69.7 ± 7.8	78.5 ± 10.8	0.027		
Daytime TBR (%)	pre-camp	2.1 [1.0;4.0]	0.038	2.3 [1.5;6.6]	1.6 [0.8;2.5]	0.86	0.507	0.009
	camp	3.2 [2.3;5.7]		4.2 [2.5;5.8]	2.9 [1.8;5.3]	0.34		
Nighttime TBR (%)	pre-camp	3.9 ± 3.2	<0.001	5.9 ± 3.2	1.9 ± 1.5	0.001	<0.001	0.017
	camp	7.6 ± 5		11.3 ± 3.9	3.9 ± 2.7	<0.001		
Daytime TAR (%)	pre-camp	41.7 ± 18.5	0.023	51.6 ± 14.2	31.9 ± 17.3	0.004	0.038	0.363
	camp	33.4 ± 13		38.0 ± 10.1	28.7 ± 14.3	0.07		
Nighttime TAR (%)	pre-camp	33.6 ± 15.9	<0.001	42.9 ± 13.5	24.3 ± 12.6	0.001	<0.001	0.011
	camp	18.3 ± 9		19.0 ± 8.1	17.6 ± 10.0	0.7		
Total daily dose	pre-camp	0.8 ± 0.2	<0.001	0.85 ± 13.5	0.74 ± 18.1	0.08	<0.001	<0.001
(TDD) (IU/kg/d)	camp	0.6 ± 0.1		0.70 ± 0.12	0.59 ± 0.13	0.027		
Percentage of	pre-camp	37.8 ± 9.7	0.040	40.4 ± 9.1	35.1 ± 9.9	0.16	0.526	0.011
basal Insulin (%)	camp	40.3 ± 8.3		41.8 ± 5.6	38.9 ± 10.4	0.40		
Daily intake of	pre-camp	198.2 ± 30.9	0.003	212.7 ± 28.4	185.0 ± 28.02	0.022	0.07	0.026
carbohydrates (g)	camp	180.2 ± 32.5		197.5 ± 27.2	166.7 ± 16.5	0.012		
Scans per day	pre-camp	13.1 ± 4.7	0.011	.	.	.	.	.
with isCGM	camp	17.7 ± 4.5		.	.	.	.	.
Difference TIR pre-camp to camp	(%)	8.0 ± 14		15.0 ± 15.9	2.8 ± 9.0	0.025	.	.
Difference TBR pre-camp to camp	(%)	2 ± 2		2.6 ± 2.8	1.9 ± 1.9	0.47	.	.
Difference TAR pre-camp to camp	(%)	−11 ± 15		−17.6 ± 16.6	−4.7 ± 9.8	0.024	.	.
Difference TDD pre-camp to camp	(%)	−19.2 ± 9.9		−17.1 ± 12.4	−19.6 ± 9.8	0.56	.	.

In case of normal distribution, data is given as mean ± standard deviation and *t*-test was uses for comparison between groups and paired-*t*-test for comparison pre-camp and camp. For quantitative data without normal distribution Mann–Whitney-U-Test and Wilcoxon Ranks Test were used for comparison and data are given as median [25th;75th percentile].

## Data Availability

The data presented in this study are available on request from the corresponding author. The data are not publicly available due to for data protection reasons.

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
