# Peer review of "Time in Range in Children with Type 1 Diabetes before and during a Diabetes Camp—A Ceiling Effect?"

_children, 2022, doi:10.3390/children9121951_

Round 1
Reviewer 1 Report
paper is well constructed but camps are not a natural setting for diabetic children and information are necessarily limited.
it would be more interesting to observe if benefic effects on glycemic control of camp could be mantained over time
however the paper is well written, even as clinical observation in limited time and in limited number of children
Author Response
Dear Reviewer 1,
We fully agree that the results of our study cannot be fully applied to the daily life of a child with type 1 diabetes. However, a camp study has clear advantages compared to at-hospital or at-home studies. On the one hand, the structural conditions are the same for all children and are easy to control, but everyday life at the camp is extremely varied and eventful, almost like in normal life. A setting in a clinic, for example, could never offer this. We would also have liked to examine how the glycemic control of camp participants continued to develop after the summer camp. However, this was not possible for us due to the retrospective design, as it was not planned in advance.
The diabetes summer camp described had the main purpose of being a recreational and educational opportunity for the children.

Reviewer 2 Report
The authors conducted a retrospective study to evaluate time in range for children with type 1 diabetes who use intermittent scanned vs real-time continuous glucose monitoring before and during summer diabetes camp.
- In the introduction, authors should specify the objectives and hypothesis of the study as well as justifying its importance.
- Authors have to discuss several limitations of their study, including the lack of control group when comparing data before and during the camp, the retrospective nature of study design, small sample size, using specific CGM systems, and how the included participants are representatives of the general population of children with T1DM.
- Authors performed bivariate analysis between children using isCGM and rtCGM (e.g, for time in range). However, they are encouraged to do multivariate analysis, to adjust for possible confounders.
- In the discussion, authors should discuss whether and how the summer camp may affect the comparison between isCGM and rtCGM.
Author Response
Dear Reviewer 2,
Thank you for your thorough review. We have implemented your suggestions and feel that our manuscript has benefited largely from them.
Ad 1.: According to your suggestions we have now described the objectives in more detail, formulated the hypotheses more clearly and also deepened the justification for our questions.
Ad 2: In the discussion, we now also point out the additional limitations you mentioned.
We emphasize once again that we can only draw limited conclusions from our study, since it was a retrospective study with a small cohort, and there was no follow-up and no non-camp control group. We also mention that one of the limiting factors was that the selection of the camp participants and the choice for the individually applied CGM systems was not made by the study team. Rather, the registration for the camp was open to all children with T1D aged 8 to 12 years and was organized by the ÖDV (Österreichische Diabetiker Vereinigung Austrian Diabetes Association). Naturally, this also limits comparability with the general population. However, comparisons with the DPV Diabetes Registry, which depicts the pediatric diabetes population in Germany and Austria very well, showed that our camp cohort was indeed comparable, especially with regard to HbA1c. This was also mentioned in the manuscript, (see line 230).
Ad 3: Due to the small sample size and the relatively heterogeneous group (many different devices, different types of insulin, different types of insulin application, different duration of diabetes, etc.), we were advised by our statistician to refrain from a multivariate analysis, as this would presumably not have yielded any further conclusive results. This would have required a much larger sample size, which is not possible due to the inert limitations of a diabetes summer camp (limited number of participants).
Ad 4: Thank you for this important suggestion, we have expanded the discussion to include considerations regarding an influence of the camp per se.
Reviewer 3 Report
The manuscript entitled “Time In Range In Children With Type 1 Diabetes Before And During A Diabetes Camp - A Ceiling Effect?” compares the time in range in children with T1D using different technologies of CGM before and after a diabetes camp. It is an interesting report, I have only found some minor issues:
1) Authors could insert a headings line on Table 3 of supplementary file, to specify that the fourth (last) column refers to the pump models.
2) On line 184, the authors mention the difference on TAR between isCGM and rtCGM. It would improve readability if the the values and p-value were also shown on the text, avoiding the need of reader to return to the table.
3) The authors find a strong association between the increase of daily scans and TIR increase, and they even compare the number of scans with previous works. Has this association been found in any previous work? Even in non-pediatric populations?
Author Response
Dear Reviewer 3
Thank you for your helpful suggestions.
Ad 1: We have inserted an additional headings line on suppl. Table 3 to specify that the fourth (last) column refers to pump models and also added a heading to indicate that the last lines refered to the various CGM devices.
Ad 2: To enhance readability we have added the values mentioned.
Ad 3: Thank you for this particularly inspiring question. We took it as an opportunity to scour the literature again. In fact, an interesting paper recently appeared in the Journal of Clinical Medicine, which was devoted to scan frequency and its impact on HbA1c. We now discuss the results of this work in our manuscript. Interestingly, the work of Leiva-Gea et. al showed that the optimum daily scan frequency might range between 15-20 scans per day. This also corresponds to the scan frequency observed at camp and we are very pleased to be able to offer another building block for further research or even regarding recommendations for the scan frequency with our manuscript.
Round 2
Reviewer 2 Report
Reviewers' comments have been adequately addressed.